# Viruses of Freshwater Mussels during Mass Mortality Events in Oregon and Washington, USA

**DOI:** 10.3390/v15081719

**Published:** 2023-08-11

**Authors:** Jordan C. Richard, Emilie Blevins, Christopher D. Dunn, Eric M. Leis, Tony L. Goldberg

**Affiliations:** 1Department of Pathobiological Sciences and Freshwater & Marine Sciences Program, University of Wisconsin-Madison, Madison, WI 53706, USA; cddunn2@wisc.edu; 2Southwestern Virginia Field Office, U.S. Fish and Wildlife Service, Abingdon, VA 24210, USA; 3Xerces Society for Invertebrate Conservation, Portland, OR 97232, USA; emilie.blevins@xerces.org; 4La Crosse Fish Health Center, Midwest Fisheries Center, U.S. Fish and Wildlife Service, Onalaska, WI 54650, USA; eric_leis@fws.gov

**Keywords:** bivalve, virome, freshwater mussel, mass mortality, die-off, Unionidae, invertebrate, biodiversity

## Abstract

Freshwater mussels (Unionida) are globally imperiled, in part due to largely unexplained mass mortality events (MMEs). While recent studies have begun to investigate the possibility that mussel MMEs in the Eastern USA may be caused by infectious diseases, mussels in the Western USA have received relatively little attention in this regard. We conducted a two-year epidemiologic investigation of the role of viruses in ongoing MMEs of the Western pearlshell (*Margaritifera falcata*) and the Western ridged mussel (*Gonidea angulata*) in the Chehalis River and Columbia River watersheds in the Western USA. We characterized viromes of mussel hemolymph from 5 locations in 2018 and 2020 using metagenomic methods and identified 557 viruses based on assembled contiguous sequences, most of which are novel. We also characterized the distribution and diversity of a previously identified mussel *Gammarhabdovirus* related to pathogenic finfish viruses. Overall, we found few consistent associations between viruses and mussel health status. Variation in mussel viromes was most strongly driven by location, with little influence from date, species, or health status, though these variables together only explained ~1/3 of variation in virome composition. Our results demonstrate that Western freshwater mussels host remarkably diverse viromes, but no single virus or combination of viruses appears to be associated with morbidity or mortality during MMEs. Our findings have implications for the conservation of imperiled freshwater mussels, including efforts to enhance natural populations through captive propagation.

## 1. Introduction

North American freshwater mussel populations play critical roles in ecosystem health and functioning [1] but have been declining severely for decades [2]. Much attention has been focused on understanding the causes of these declines in the species-rich mussel populations of the Eastern United States. Despite myriad known stressors (habitat destruction, water quality declines, invasive species, etc.) affecting all North American freshwater mussel populations [2], much remains unknown about the causes and consequences of their declines [3]. For example, in a systematic assessment of 124 published studies providing conclusions about the causes of mussel population losses, Downing et al. [4] found that 75% of studies suggested multiple potential causes and only 48% of studies provided plausible links between extirpations and causes. While pollution/water quality issues and habitat destruction/alteration were the most frequently cited causes, a total of 17 major causes were described. Notably absent from those listed is the possibility of infectious disease, which is a major driver of biodiversity losses in other taxa (e.g., white nose syndrome in bats [5], amphibian Chytridiomycosis [6]) and has been a subject of increasing research interest for freshwater mussels [3,7].

Freshwater mussel mass mortality events (MMEs), in which a population experiences an acute die-off affecting a significant proportion of the population, are a particularly enigmatic subset of mussel population declines. While some MMEs result from obvious stressors (e.g., drought, flood, and toxic contaminant spills), most lack clear causes. In such cases, infectious disease has long been suspected of playing a role [3,8], though few studies have explored this possibility in detail. Recent work has found associations between freshwater mussel MMEs and viruses [9,10] and bacteria [11,12,13] in Eastern US populations, and has also revealed the presence of parasites in declining mussel populations [14,15]. 

An infectious disease often plays a primary role in marine invertebrate MMEs [16,17], though it is well-documented that such diseases often trigger mortality events as part of a larger set of interconnected factors, which also include abiotic and anthropogenic stressors [18]. In marine oysters, for example, Ostreid herpesvirus 1 (OsHV-1; *Malacoherpesviridae*) often causes mortality rates up to 100% in larvae and juveniles associated with changes in host metabolism and loss of energy reserves [19]. Secondary bacterial infection by *Vibrio* spp. and environmental factors (particularly, temperature, and salinity) strongly influence OsHV-1 mortality rates [20,21]. In shrimp aquaculture, viral disease outbreaks are a major challenge and cause economic losses worldwide [22,23]. Penaeid shrimp viruses such as white spot syndrome virus (WSSV; *Nimaviridae*), infectious hypodermal and hematopoietic necrosis virus (IHHNV; *Parvoviridae*), and infectious myonecrosis virus (IMNV; *Totiviridae*) have therefore been studied in detail. While evidence from recent freshwater mussel MME investigations suggests that infectious diseases (in particular, viruses and bacteria) may play a contributing role [9,12], the majority of this research has been based on studies of populations in the Eastern US.

Western freshwater mussel populations from the Pacific Region are similarly in decline, but they have received less attention, even though Western USA freshwater mussels play the same critical roles in ecosystem functioning and face many of the same anthropogenically driven population declines as their Eastern USA counterparts [24]. Recent MMEs of Western mussel populations have been recorded in numerous rivers throughout the US states of Washington, Oregon, and Idaho, including the Chehalis, Crooked, Middle Fork John Day, Grand Ronde, John Day, Weiser, and Owyhee Rivers [25]. As with MMEs involving Eastern US mussels, infectious disease has often been hypothesized as a driver of Western US mussel MMEs, but to our knowledge, to date, no formal investigations of this hypothesis have been conducted in the Western USA.

Here, we describe the results of a two-year epidemiologic study to characterize the role of viruses in Western mussel MMEs in OR and WA, USA. By pairing field data on mussels during MMEs and at control sites with metagenomic data on mussel viromes, we examine whether particular viruses or combinations of viruses are associated with the health status of mussels. Our results reveal a large number of viruses in Western mussel populations but, unlike in many Eastern US populations, few associations between viruses and MMEs. Also, we previously identified in *M. falcata* a novel gammarhabdovirus (*Rhabdoviridae*: *Gammarhabdovirinae,*) chemarfal virus 1 (CHMFV-1), closely related to pathogenic finfish viruses in the genus *Novirhabdovirus*, which are of global concern for wild fisheries and aquaculture [26]. We describe additional data on this virus, including its distribution, diversity, and association with disease in mussels.

## 2. Materials and Methods

### 2.1. Study Sites 

#### 2.1.1. Columbia River Watershed, OR

The Crooked River is a tributary of the Deschutes River, which is a tributary of the Columbia River, in Oregon, USA. Biologists from the Pacific Northwest Native Freshwater Mussel Workgroup first reported observation of *G. angulata* mass mortality in the Crooked River in 2014 from a site (CRO) within Smith Rock State Park near Terrebonne, OR (Figure 1) [25]. The initial report described thousands of recently dead shells as well as many mussels that were still alive. We documented freshly-dead shells, moribund, and apparently healthy mussels annually from 2018 to 2022, indicating the MME remains ongoing but has not yet eliminated the population. We sampled *G. angulata* from the Crooked River in Smith Rock State Park, Oregon (CRO) in 2018 and 2020 (Figure 1). *G. angulata* are the dominant species and were observed in large numbers in both moribund and apparently healthy conditions, while the relatively less abundant *Anodonta* spp. were only observed in apparently healthy conditions. Moribund mussels were laying on the substrate surface with shells gaping open and were slow to respond and close their shells when soft tissues were stimulated. Apparently healthy mussels were identified as those that remained buried in the substrate, responded quickly and strongly to tactile stimulation, and showed no obvious signs of abnormal behavior.

The Owyhee River is a tributary of the Snake River, in Nevada, Idaho, and Oregon. In 2019 the Xerces Society for Invertebrate Conservation and Oregon state fish biologists surveyed the Owyhee River in response to a public report of numerous recently dead shells. The survey found unburied mussels and many recently dead shells of *G. angulata* throughout the ~49 river miles (RM) surveyed. Live *G. angulata* were also present throughout the survey area. 

#### 2.1.2. Chehalis River Watershed, WA

The Chehalis River originates in Southwestern WA and drains into Gray’s Harbor and the Pacific Ocean. A Washington state fish biologist first reported a large-scale mussel MME in the Chehalis River near Oakville, WA in 2015. Follow-up surveys in 2017–2020 confirmed mass mortality of *M. falcata*, *G. angulata*, and *Anodonta oregonensis* from RM 43 to RM 24. Additional resurveys in 2019 at several sites (RM 24 and RM 33) did not successfully relocate any live *G. angulata*, despite having observed 21 live animals at RM 24 and 5 live animals at RM 33 in 2017. Surveys in 2017 at RM 76 documented a large (>100,000 individuals) bed of *M. falcata*, but resurveying at RM 76 in 2019 revealed evidence of substantial die-off, with an estimated 42% mortality rate of *M. falcata*. A small number of live and dead *G. angulata* were also observed in 2020 but were not observed at all during a 2022 survey. 

The Skookumchuck River is a tributary of the Chehalis River in WA. While an ongoing mussel MME has been documented in the Chehalis River above and below the Skookumchuck’s confluence with the Chehalis River, we have not observed any unusual levels of mussel mortality in the Skookumchuck. However, in 2022 in another tributary, the Newaukum River, Xerces Society biologists investigated a MME impacting a large *M. falcata* bed.

### 2.2. Sample Collection

#### 2.2.1. Western Ridged Mussel (*Gonidea angulata*) Sampling

On 8 October 2018, we collected samples from 5 moribund and 5 apparently healthy *G. angulata* in the Crooked River from the Smith Rock State Park site (CRO). Mussels were collected whole, wrapped in damp paper towels, and placed in coolers over wet ice with barriers to keep them chilled without freezing. Samples were shipped overnight to a laboratory for processing, where we extracted hemolymph from the anterior adductor muscle sinus, as previously described [9]. Samples were stored immediately in −80 °C freezers until further processing. We also opportunistically collected samples from 3 moribund *G. angulata* during a MME in the Owyhee River (OWY) in July 2018 using the same methods. 

We returned to the same Crooked River sample site (CRO) on 4 September 2020 and collected an additional 5 moribund and 6 apparently healthy *G. angulata*. We also sampled 3 apparently healthy *G. angulata* in the Chehalis River in August 2020 during sampling of the Chehalis River MME site LCM1 described below. All 2020 samples were collected non-lethally in the field using the same method of extracting hemolymph from the anterior adductor muscle sinus. Hemolymph samples were stored immediately in microcentrifuge vials on dry ice in the field before overnight shipping and transfer to −80 °C freezers in the laboratory where they were preserved until further processing. A summary of samples is provided in Table 1 and full sample details are in Appendix A.

#### 2.2.2. Western Pearlshell (*Margaritifera falcata*) Sampling

We sampled 6 moribund *M. falcata* from a MME site at RM44 (LCM2) in the Chehalis River on 26 September 2018 near Oakville, OR. On the same day, we sampled 5 apparently healthy *M. falcata* from the nearby Skookumchuck River at RM2.5 (SKO). On 26 August 2020, we sampled a MME site in the Chehalis River (LCM1) just upstream of the confluence with the Newaukum River near RM75.5, collecting samples from 12 moribund and 13 apparently healthy *M. falcata* (Figure 1). On 24 August 2020, we sampled an additional 14 apparently healthy *M. falcata* from the Skookumchuck River SKO site we visited in 2018. Samples from 2018 were collected as whole mussels and processed in the laboratory, while 2020 samples were collected non-lethally in the field, as described above for *G. angulata*. 

### 2.3. Viral Nucleic Acid Extraction and Sequencing

We prepared mussel hemolymph samples for virus sequencing using previously described methods [27]. Briefly, we first centrifuged hemolymph at 10,000× *g* for 10 min to pellet large non-viral particles. We then transferred the supernatant to new vials, and concentrated virus particles via centrifugation at 25,000× *g* for 3 h. We used the QIAamp MinElute Virus Spin Kit (Qiagen, Hilden, Germany) to extract total nucleic acids and converted RNA to double-stranded cDNA using the Superscript IV system (Thermo Fisher, Waltham, MA, USA) with random hexamers and NEBNext Ultra II Non-Direction RNA Second Strand Synthesis Module. We prepared DNA libraries using the Nextera XT DNA Library Preparation Kit (Illumina, San Diego, CA, USA). We sequenced libraries using a MiSeq instrument.Samples from 2018 were sequenced with V3 chemistry, 600 cycle kit, while 2020 samples were sequenced with V2 chemistry, 300 cycle kit (Illumina, San Diego, CA, USA). Samples were sequenced on a total of 7 separate runs. 

### 2.4. Bioinformatics and Virus Classification

We demultiplexed reads and quality-trimmed them to ≥Q30 using CLC Genomics Workbench v23.0.2 (Qiagen, Hilden, Germany) and we discarded short reads (length < 50 nt). We then used public [28] and in-house databases to remove reads mapping with ≥70% nt similarity over ≥80% of read length to low-complexity regions, eukaryotic sequences, ribosomal sequences, and known laboratory contaminants. For each individual sample, we assembled the remaining reads into contiguous sequences (contigs) using metaSPAdes v3.15.2 [29]. We used CD-HIT-EST [30] to cluster contigs from all individuals with >97% nt similarity and return the longest available representative sequence for each, as previously described [10]. Resulting clustered contigs >1000 nt were queried against sequences in GenBank using both nucleotide (BLASTn) and deduced amino acid (BLASTx) searches of the nt and nr databases, respectively. We retained all contigs matching virus sequences with E-values < 10^−20^ to minimize false positive identifications if they did not have equivalent or better matches to eukaryotic, bacterial, or bacteriophage sequences. We annotated putative viral contigs using Cenote-Taker 2 in annotation mode [31] and the Conserved Domain Database [32]. We removed putative viral contigs matching known laboratory contaminants [33,34].

For identified virus contigs, we estimated taxonomy using the DIAMOND BLASTx + MEGAN pipeline [35,36] with default parameters to identify viruses based on a LCA approach using the top 10 closest matches in the NCBI nr database. When unclassified contigs contained conserved domains corresponding to particular virus groups, we assigned them to lower taxonomic levels if those matches were consistent with BLAST results and that virus group’s typical genome organization. For example, unclassified circular contigs from the MEGAN pipeline containing Rep protein domain matches and putative capsid proteins typical of CRESS viruses were assigned manually to Phylum Cressdnaviricota. We used the results of these updated taxonomic rankings to visualize the virus diversity in the dataset with KronaTools [37]. We assigned virus names in a manner consistent with previous methods in [10,26] using names based on the sampling site, genus, and species of the samples (e.g., chemarfal virus 2 = Chehalis *Margaritifera falcata*). To avoid confusion in future taxonomic revisions, we did not include taxonomic descriptors in virus names, as per published guidelines [38]. For viruses matching previously-identified viruses, we used names previously established in GenBank records. Our threshold for using previously established names was ≥90% nt identity over the RNA-dependent RNA polymerase for RNA viruses and over the Rep protein for circular DNA viruses. 

### 2.5. Statistical Analysis

We mapped trimmed, quality-filtered reads >50 nt from each individual to all identified virus contigs with a stringency of ≥95% nt identity over ≥95% read length using the “map reads to reference” function in CLC Genomics Workbench. We counted a virus as present in a sample if it contained ≥2 reads mapping to a contig at the specified thresholds. We chose the ≥2 read threshold to avoid counting viruses as present in a sample if they contained only a single mapped read in which the corresponding paired read was discarded due to quality filtering. Read mapping results were used to calculate a measure of viral read abundance normalized by contig length to account for differing target sequence lengths for each virus [39]. The resulting measure was normalized to reads per million to account for differing sequence depths across individuals and reported as viral reads per million per kilobase of target sequence (vRPM/kb, hereafter, “virus abundance”). We also calculated the total number of viruses present in each sample (hereafter, “virus richness”). We visualized the distribution of viruses within each sample using the taxonomic rankings we assigned to each contig and the mapped read counts by importing the data to *Phyloseq* [40] in R and creating relative abundance plots for all individuals.

To screen for viruses associated with mortality in each population, we first separated the datasets by species. We used an epidemiologic case–control approach, in which moribund mussels were classified as cases and apparently healthy mussels were classified as controls. Cases were characterized by mussels found to be unburrowed and lying atop the river bottom, often adjacent to other normal burrowed and open, filtering mussels. These mussels tended to respond only slowly when touched, if at all, and appeared to gape slightly open without actively filtering. Controls were observed to be securely burrowed into the river substrate and reactive, closing when handled. For the *G. angulata* dataset, we used 2-tailed *t*-tests to evaluate whether individual viruses were present at higher abundance in case vs. control mussels for both the 2018 and 2020 samples. We then used ANOVA to determine if virus richness differed between any of the case vs. control groups for each year in the Crooked River or the *G. angulata* group collected from the Chehalis River. We also used a machine learning approach to assess the potential importance of individual viruses in distinguishing case vs. control mussels in the *G. angulata* dataset. We used the function randomForest (settings: ntree = 50,000, mtry = 19, replace = FALSE) followed by the importance function of the randomForest package [41] in R version 4.2.2 [42] to model the relationship between viral read depths for each virus and individual health status. The mean decrease in Gini values was used to rank individual viruses based on their relative contribution to distinguishing between case vs. control mussels.

For the *M. falcata* dataset, we first used 2-tailed *t*-tests as a screening tool to evaluate whether individual virus abundances were significantly different between (1) case vs. control *M. falcata* sampled in the Chehalis River in 2020; (2) case Chehalis River *M. falcata* and control Skookumchuck River *M. falcata* in 2020; and (3) case Chehalis River *M. falcata* and control Skookumchuck River *M. falcata* in 2018. We also used ANOVA to assess whether virus richness differed significantly between mussel samples from the following groups: 2018 control Skookumchuck River; 2020 control Skookumchuck River; 2018 case Chehalis River; 2020 control Chehalis River; and 2020 case Chehalis River. As with the *G. angulata* dataset, we used random forests (settings: ntree = 50,000, mtry = 20, replace = FALSE) to model the relationship between the abundance of each virus and mussel health status.

We used a multivariate approach to assess the complete dataset of all sites, health conditions, species, and years to explore factors influencing the viromes of all mussels in the study. We used nonmetric multidimensional scaling (NMDS) implemented via the *metaMDS* function (settings: distance = “bray”; center = TRUE; scale = TRUE, *n* = 1000) of the *vegan* package [43] in R to visualize the relationships between samples from the various sites, species, and health conditions. Finally, we used variation partitioning (*varpart* function of the *vegan* package in R) to explore the relative influence and proportion of virome variation that could be attributed to collection year, sample site, species, and health status.

To investigate the diversity of chemarfal virus 1 within and between populations, we conducted an analysis of the rhabdovirus L gene, consistent with phylogenetic methods designated for the *Rhabdoviridae* family by the International Committee on Taxonomy of Viruses [44]. For all samples with reads mapping to any portion of the virus genome, we re-mapped quality-screened (i.e., trimmed, quality-filtered, and contaminant-screened) reads to the chemarfal virus 1 L gene (GenBank ID OQ368744.1) using thresholds of 80% similarity over 70% of read length. Sequences were hand-aligned, as there were no indels. From the aligned sequences, we used MEGAX (v10.2.6 [45]) to calculate average pairwise nt differences among the 5934-nt segments using 100 bootstrap replicates to estimate the standard error. In order to investigate the relationship between the previously identified rhabdovirus chemarfal virus 1 and mussel mortality, we examined patterns of association between cases and controls as described above for other viruses. 

## 3. Results

### 3.1. Virus Sequencing, Characterization, and Diversity

We sequenced hemolymph samples from 77 mussels (27 *G. angulata*, 50 *M. falcata*) from 5 sample sites in 2018–2020. After quality trimming and length filtering, the average sequence depth was 2,065,369 (standard deviation (SD) 1,103,334) reads per individual. From these reads, we identified 557 viral contigs ranging from 1016–16,281 nt (Appendix A). A small number (6.8%, n = 38) of contigs matched known viruses with BLASTn results showing >90% nt similarity. Several of these were nearly identical matches to viruses we identified in previous studies of eastern freshwater mussel mortality events [9,10], including flactilig virus 7 (dicistrovirus), flactilig virus 1 (nodavirus), crogonang virus 133 (a near-complete calicivirus genome matching the partial genome of clinch calicivirus 1), and chemarfal virus 122 (a near-complete dicistrovirus genome matching the partial genome of clictilig virus 2). Another 16 viruses met our thresholds for using previously-established virus names (i.e., ≥90% nt identity across the complete RNA-dependent RNA polymerase for RNA viruses and across the Rep protein for circular DNA viruses), most of which were members of the Picornavirales.

The majority of contigs were only distantly related to known viruses. Initial taxonomic assignments based on the DIAMOND + MEGAN pipeline placed 19.2% (n = 107) of contigs into the “not assigned” group. Of the remaining contigs, the lowest levels of classification placed 11.1% (n = 62) as “Viruses”, 14.7% (n = 82) as “Riboviria” and 18.3% (n = 102) as unclassified viruses most closely related to those of Shi et al. [46] (Appendix A). Our further examination of contigs identified with low taxonomic resolution allowed us to place a large number of contigs into lower taxonomic levels based on patterns of conserved domains, genome structure, and nearest BLAST matches. This revised estimate of taxonomic position for the full dataset revealed that the largest fractions of the virome comprised members of the (+)ssRNA Picornavirales (35% of all contigs). Of these, the largest proportion (44% of the Picornavirales and 15% of all contigs) were unclassified below the Order level, followed by 33% (11% of all contigs) in the *Marnaviridae*, 18% (6% of all contigs) in the *Dicistroviridae*, and the remaining 5% of Picornavirales distributed among the *Picornaviridae*, *Iflaviridae*, *Caliciviridae*, *Noraviridae*, and *Polycipiviridae*. Next, 20% of all contigs were placed within the ssDNA Shotokuvirae, 93% of which were unclassified members of the Cressdnaviricota (Figure 2). Other taxonomic assignments placed ≥1% of all contigs in the families *Nodaviridae* (6%), *Tombusviridae* (6%), *Picobirnaviridae* (3%), *Circoviridae* (2%), *Solemoviridae* (2%), *Astroviridae* (1%), *Botourmiviridae* (1%), and *Parvoviridae* (1%). These revised taxonomies allowed us to confirm all 557 contigs as viruses and reduced the number of contigs listed as “viruses, unknown” to 8% of the full dataset. 7% of the dataset remained as “Riboviria, unclassified” and 1% of contigs were assigned to the unclassified Cruciviruses-like group. 

A total of 67.1% (n = 374) of viruses were found in *G. angulata*, with 21.9% (n = 122) of those exclusively observed in *G. angulata*; 78.1% (n = 435) of viruses were found in *M. falcata*, with 32.9% (n = 183) of those unique to *M. falcata*; 45.2% (n = 252) of viruses were observed in at least 1 individual of both species; 32.0% (n = 178) of viruses were found exclusively from a single study site, while 68.4% (n = 381) of viruses were found in two or more sites. By year of sampling, 71.3% (n = 397) of viruses were observed in 2018, while 91.2% (n = 508) were observed in 2020; 62.5% (n = 348) of viruses were observed in both years.

Analysis of the chemarfal virus 1 L gene found 12 individuals with complete coverage, 1 of which was the sample originally used to characterize the virus [26]. The average (±SE) nt difference for all 12 sequences was 16.33 nt (0.275%) ± 2.35 nt (0.040%). 

### 3.2. Associations between Viruses and Health Status

We found only three viruses that differed significantly in their viral read counts between case and control *G. angulata* in the 2018 Crooked River samples. Chemarfal virus 122 (a Dicistrovirus; *Dicistroviridae*) and chemarfal virus 1 were found in all 5 cases and in none of the 5 control animals (*p =* 0.002 and *p =* 0.004, respectively). Chemarfal virus 122 was a nearly complete genome with 98.5% nt sequence similarity to clictilig virus 2, a partial dicistrovirus genome we previously identified in [10]. Skomarfal virus 2 (a Tombus-like virus most closely related to two freshwater mussel viruses we previously described in [10]) read counts were significantly higher in the controls than in the cases (*p =* 0.029). None of these viruses were observed in any individuals from the 2018 Owyhee River cases, 2020 Crooked River controls, or 2020 Crooked River cases. Chemarfal virus 122 was observed in all three 2020 LCM1 controls, but neither of the other two viruses was observed in these samples. Both chemarfal virus 122 and chemarfal virus 1 were observed in numerous *M. falcata* samples from both 2018 and 2020. Only 1 virus (crogonang virus 158, an unclassified virus most closely related to members of the *Solemoviridae*) had significantly different viral read depths between cases and controls in the 2020 Crooked River samples, with higher levels observed in the controls (*p =* 0.038). Crogonang virus 158 was not observed in any other *G. angulata* samples in 2018 or 2020, though it was present in a single *M. falcata* sample from the 2020 LCM1 dataset; 16 viruses were observed exclusively in the 3 individuals sampled from the Owyhee River. All vRPM/kb values for individual viruses and *G. angulata* samples are listed in Appendix A. Overall mean (±SD) virus richness for *G. angulata* was 80.7 ± 50.9 and did not differ between any of the sample groups (ANOVA *p* = 0.864) (Figure 3). Random forest models built to assess differences between *G. angulata* cases and controls based on virus abundance data resulted in an overall out of bag (OOB) estimate of error rate of 40.7%, with classification error rates for cases at 38.5% and controls at 42.9%. Variable importance rankings revealed that crogonang virus 1 had a much higher contribution to the model, followed by crogonang virus 117, crogonang virus 159, and crogonang virus 6. All other viruses—including chemarfal virus 1 (ranked as 8th most important)—had relatively lower contributions to distinguishing cases from controls and showed a consistent decrease in importance (Figure 4). 

For the 3 *G. angulata* cases collected from the Owyhee River (samples OGH01 through OGH03), we observed 40, 55, and 72 viruses in each individual. We found 83 viruses total from the OWY site, 16 of which were never observed in any other samples from the study. Samples OGH01 and OGH02 had similar viral read distributions, primarily driven by picorna-like viruses of the Pisuviricota followed by the tombus-like and noda-like viruses of the Kitrinoviricota. OGH03 differed from the other two, with the largest proportion (74%) of relative viral reads attributed to ssDNA viruses of the Cressdnaviricota.

In the *M. falcata* dataset, nine viruses had significantly different viral read depths between cases and controls in the 2020 LCM1 samples. Two of these, flactilig virus 1, *Nodaviridae* and flactilig virus 7, *Picornaviridae*, had significantly higher levels in the control group, while seven had higher levels in cases. Both flactilig virus 1 and flactilig virus 7 were found at 100% prevalence in LCM1 controls and 0% prevalence in LCM1 cases and were never observed in *G. angulata* (which include individuals collected contemporaneously with the LCM1 *M. falcata*). When comparing 2020 *M. falcata* viruses between pooled LCM1 samples (i.e., all cases and controls) vs. 2020 SKO controls, 183 viruses were significantly different between the 2 groups, with 59 at higher levels in the SKO controls and 124 higher in the LCM1 cases. For the 2018 LCM2 cases vs. 2018 SKO controls, 25 viruses were significantly different, with 11 found at higher levels in the SKO controls and 14 at higher levels in the LCM2 cases. Five viruses were significantly elevated in Chehalis River (LCM2 in 2018, LCM1 in 2020) cases vs. SKO controls in both study years: chemarfal virus 122 (*Dicistroviridae*), chemarfal virus 265 (*Picornaviridae*), chemarfal virus 24 (*Marnaviridae*), chemarfal virus 194 (Cressdnaviricota), and chemarfal virus 273 (*Dicistroviridae*). None of these 5 viruses differed significantly between 2020 cases and controls at the LCM1 site. Overall mean (±SD) virus richness for *M. falcata* was 102.3 ± 78.3 and did not differ between any of the sample groups (ANOVA *p =* 0.064) (Figure 3). Chemarfal virus 1 was significantly higher in 2018 LCM2 cases than in 2018 SKO controls (*p* = 0.02), with a prevalence of 100% in cases (n = 6) versus 40% in controls (n = 5). However, this association was not observed in subsequent 2020 sampling, where average virus abundance in SKO controls was slightly higher than—but not significantly different from—that of LCM1 cases (*p* = 0.53). Similarly, the chemarfal virus 1 level did not differ significantly between LCM1 cases and controls collected from the same site (*p* = 0.27). All vRPM/kb values for individual viruses and *M. falcata* samples are listed in Appendix A. Random forest models assessing differences between *M. falcata* cases and controls based on virus abundance data performed better than those of the *G. angulata* models, with an OOB estimate of an error rate of 14%. Classification error for the *M. falcata* random forest model was 27.8% for cases and 6.3% for controls. Variable importance rankings showed that flactilig virus 1 and chemarfal virus 122 were more important than other viruses. Chemarfal virus 265, crogonang virus 6, and flactilig virus 7 were slightly more important than other viruses, and the remaining viruses had relatively even importance values that steadily decreased throughout the dataset (Figure 4).

For both species, the distribution of relative viral read abundances among phyla was generally consistent with their observed richness. For example, the Pisuviricota contained 41% of the 557 contigs we observed and averaged 38% of relative read abundances across all individuals in the study (Figure 5). The Cressdnaviricota comprised 19% of all contigs identified and an average relative read abundance of 24%, followed by the Kitrinoviricota with 17% of contigs and 15% of average relative reads. The Negarnaviricota created a prominent exception to this pattern. While we found only 1 virus in this phylum (the rhabdovirus chemarfal virus 1), it represented 13% of average relative reads across all individuals. Nine individuals had ≥67% of their viral reads attributed to chemarfal virus 1, with a maximum of 99% of viral reads in 1 individual (MSK13, a *M. falcata* control from SKO in 2020).

### 3.3. Multivariate Virome Assessment

NMDS ordination and subsequent visualization of the virome data for all samples are shown in Figure 6. NMDS axis 1 represents differences in viromes based on sites and mussel species (which are largely confounded given the sampling design; only one site yielded samples of both species). NMDS axis 2 represented the variability of viromes within a given species and site. We saw little effect of either case versus control samples or sampling year in the ordination results. Variation partitioning of the total virome dataset among fractions related to sample year, sample site, species, and health status yielded a model with an adjusted *R*^2^ value of 0.31. Individual *R*^2^*Adj* values and *R*^2^*Adj* values uniquely attributable to individual variables (in parentheses) were as follows: sample site = 0.283 (0.131), species = 0.128 (0.006), sample year = 0.052 (0.013), and health status = 0.024 (0.008). The sample site was the only variable with a large proportion of unique variation explained (13.1%), as most of the 12.8% of variation explained by species occurred in combination with the sample site (10.2/12.8%). Proportions of unique and shared variation explained by each component are shown in Figure 7.

## 4. Discussion

We identified 557 virus contigs from 77 mussels sampled at 5 sites throughout several major Northwestern US watersheds in 2018–2020. The majority of these viruses are uncharacterized (+)ssRNA viruses and are distantly related to viruses found in previous studies of invertebrates (including freshwater mussels) [46] and aquatic environmental samples [47]. Several viruses we identified were previously found in other systems. Flactilig virus 1 and flactilig virus 7 were observed in multiple populations of *M. falcata* and showed significant associations with the control mussels. Both of these viruses were previously identified at high prevalence in multiple populations of mucket (*Actinonaias ligamentina*) from the Midwestern US across 3 consecutive years, suggesting that they may represent common members of a core virome for some mussel species. We found Western US mussels had high virus richness in comparison to our previous investigations of viruses in MMEs of Eastern US mussels. For example, in a study of 29 muckets (*A. ligamentina*) from three watersheds in the Midwestern and Eastern US over several years, we found 38 viruses using similar methods [10]. In that study, we found that average virus richness varied by population and health status, but was generally low, with a maximum of 15.75 viruses per mussel during an active mortality event. While we found no differences in virus richness between groups in the present study, we note that the mean (±SD) virus richness for *M. falcata* was 102.3 ± 78.3, and 80.7 ± 50.9 for *G. angulata*. While differences in sequencing depth, number of study sites and species, and other factors may partially account for some of the differences observed in virus richness between studies, it is possible that Western US mussels harbor higher viral diversity than mussels of the Eastern US.

Our initial results from 2018 suggested several potential viruses were associated with cases for both *G. angulata* and *M. falcata*. In particular, chemarfal virus 1 was significantly higher in 2018 cases than controls for both species. While chemarfal virus 1 was present at very high levels in many samples, our 2020 samples found it in a number of controls from MME sites as well as controls from sites with no mortality observed. Indeed, in the full study, we found no viruses that were significantly elevated in *G. angulata* cases vs. controls across both years. Similarly, for the *M. falcata* data, no viruses were consistently significantly elevated in cases vs. controls across all years and sites. Collectively, our statistical analyses of the two species’ viromes found few differences between cases and controls. Virus richness did not differ between any of the sample groups in the study. For both mussel species, random forest misclassifications for cases were relatively high (38.5% for *G. angulata* and 27.8% for *M. falcata*), though classification was much more accurate for controls in the *M. falcata* dataset (6.3%) than in the *G. angulata* dataset (42.9%). This is likely due to sampling design, as the *M. falcata* dataset included 14 samples from a separate control site (the Skookumchuck River). When comparing the results of cooccurring cases and controls, we found no consistent differences in virus abundance that suggested a potential etiologic role for viruses in the observed MMEs.

Our results show that the rhabdovirus chemarfal virus 1 (CHMFV-1) is not likely associated with disease and regularly occurs at high levels in individuals and populations exhibiting no outward signs of morbidity or mortality. This observation concords with previous data on rhabdoviruses in invertebrates. Many rhabdoviruses are vector-borne, with arthropods serving as vectors that transmit the viruses to animals or plants [44]. Although the viruses can cause severe disease in their ultimate host, their consequences for the invertebrate vector appear to be minimal. For example, *Vesiculovirus Indiana* (vesicular stomatitis Indiana virus) is a well-studied virus that causes disease in livestock and wild mammals but causes little or no cell death in its persistently infected insect vectors [48]. Similarly, many rhabdoviruses cause severe agricultural losses in plants while causing no significant cytopathology in the central nervous system of their insect vectors, where they establish persistent infections [49]. The low genetic diversity we documented among CHMFV-1 L gene sequences is consistent with a study of mosquito rhabdoviruses, where L genes were nearly perfectly conserved, while N genes were much more variable [50].

We found that approximately 1/3 of the variation in mussel virome composition could be explained by species, site, date of sampling, and health status (case versus control). The majority of this variation was explained by location (sampling site). When accounting for the effects of other variables, the variation uniquely attributable to time, species, and health status totaled only 2.6%. Many large-scale studies of viromes are based on cross-sectional surveys at a specific place and time with little ecological context [51], but this is changing. For example, Raghwani et al. [52] found that viruses in the family *Picornaviridae* in wild rodents exhibited fluctuations in richness and evenness over time that were associated with changes in host density, ambient temperature, rainfall, and humidity. Campbell et al. [53] found significant differences between the viromes of red foxes from urban and rural settings, highlighting the importance of anthropogenic influences on wildlife viromes. Similarly, Campbell et al. [54] found that viromes of Atlantic bonefish (*Albula vulpes*) differed across areas of the Caribbean, with viral richness and load highest in the most anthropogenically impacted locations. In a study of the cave-dwelling freshwater mussel *Congeria kusceri* from the Balkan Peninsula, Scapolatiello et al. [55] found a pronounced increase in the abundance of all viruses detected during the summer season, suggesting that seasonality may play an important role in shaping mussel viromes. Surrounding landscape factors, such as wastewater discharges, may also shape aquatic viromes, as recently found in [56]. While we included sampling date as a variable in our models, all of our samples were collected during a relatively short period during the summer months. Future work characterizing Western US mussel viromes over time could be informative for understanding whether the high virus richness we observed remains stable throughout the year. Identifying the root causes of mussel declines, in general, would benefit from additional sources of data such as water chemistry analyses [57], histopathologic analyses [58], metabolomics [59], and analyses of other classes of microbes [11]. 

Emerging viruses have been implicated in wild and captive animal mortality events worldwide. Mass mortality events due to infectious disease in shrimp aquaculture cost USD ~1 billion in losses annually [22] and are largely caused by viral pathogens like white spot syndrome virus (WSSV), infectious hypodermal and hematopoietic necrosis virus (IHHNV), and Penaeid shrimp myonecrosis virus (PsIMNV) [23]. Similarly, cyclic MMEs of Pacific oysters in more than 12 countries are regularly attributed to *Ostreid herpesvirus 1* (OsHV-1) and its variants [17]. These outbreaks appear to be caused by complex interactions of factors, including OsHV-1, bacteria, parasites, and environmental stressors [60]. Indeed, in previous studies, we observed significant associations between MMEs of Eastern US mussels and viruses [9,10] and bacteria [11,12], with environmental factors likely contributing. However, our data from Western US mussel MMEs illustrate that mussel MMEs are not universally associated with viruses, at least within the limits of our detection capabilities. In this light, we note that our statistical power may also have been reduced by asynchronous sampling of sites, time lags between etiological factors affecting mussels and our sampling of observed mortality, or the sheer number of viruses we detected.

Rapid population declines and the growing list of extinct freshwater mussels represent a major global conservation crisis. Population augmentation via captive propagation and/or wild animal translocations is the primary tool for preventing extinctions and restoring populations [61]. The need for freshwater mussel conservation hatcheries continues to grow amidst a backdrop of unexplained MMEs, population declines, and extinctions worldwide. It is critical to support these conservation efforts with effective information characterizing the role of pathogenic, persistent, and harmless viral infections. While identifying pathogenic infections is important in the rearing of both commercially and ecologically important species, for highly-imperiled freshwater mussels, misidentifying a virus as harmful could lead to the unnecessary culling of progeny or restrictions on natural population augmentations. Effectively characterizing viruses in these globally imperiled animals is therefore important not only for preventing the unintentional spread of pathogens, but also for identifying viruses that do not warrant biosecurity concerns.

## Figures and Tables

**Figure 1 viruses-15-01719-f001:**
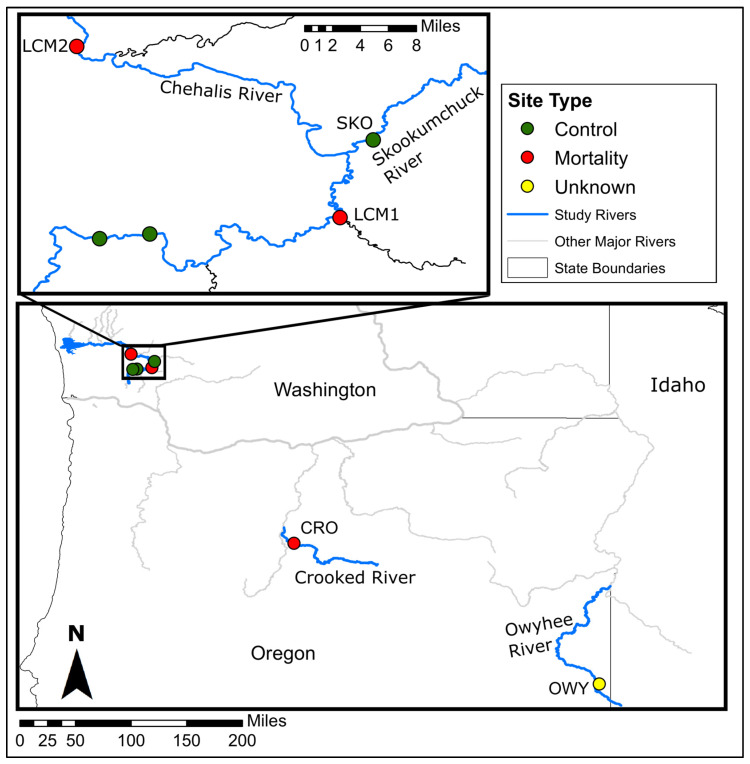
Map of sampling sites and monitoring locations used in this study. Inset map shows more detail for sites in the Chehalis River watershed. Sample sites are labeled with their corresponding codes as described in the text. The two unlabeled sites on the Chehalis River show sites where we confirmed that mussel mortality was not occurring, but did not sample for this study.

**Figure 2 viruses-15-01719-f002:**
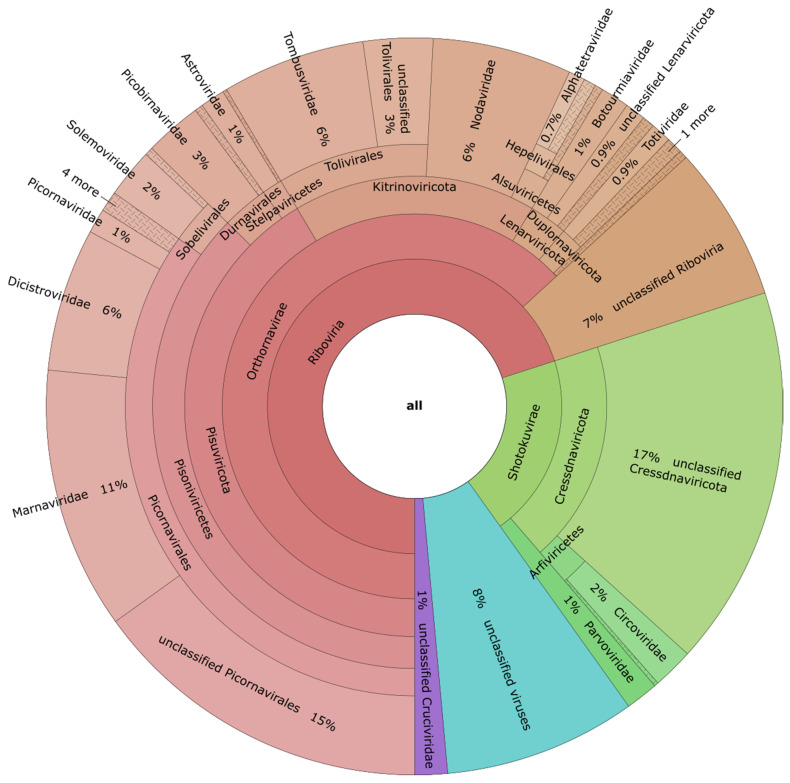
KRONA diagram showing the revised taxonomy assignments produced using MEGAN followed by identification of conserved domains for the 557 viruses found in this study. The majority of virus contigs were most closely related to (+)ssRNA viruses, particularly members of the Picornavirales, as well as tombus-like and noda-like viruses. There were also a number of ssDNA viruses, including CRESS viruses and members of the *Parvoviridae*. Groups representing <0.7% of all viruses are collapsed and depicted with cross-hatch shading.

**Figure 3 viruses-15-01719-f003:**
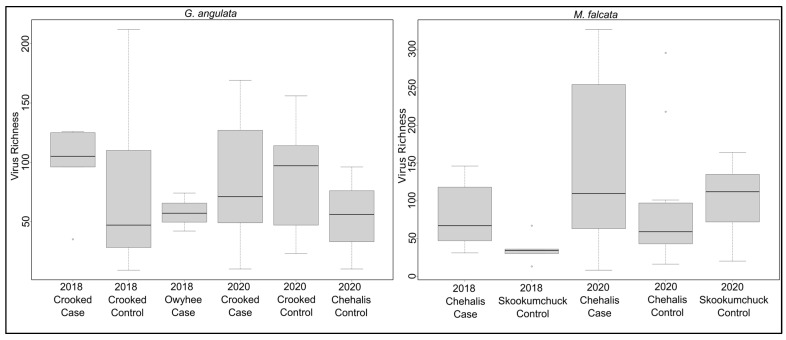
Boxplots of virus richness observed in sample groups of *G. angulata* (**left**) and *M. falcata* (**right**). ANOVA results indicated that there were no significant differences in virus richness for any sample group within either species’ dataset.

**Figure 4 viruses-15-01719-f004:**
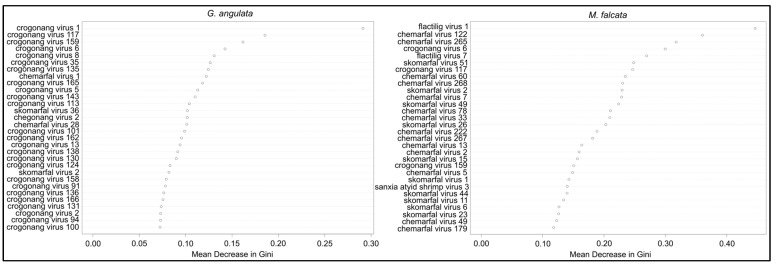
Random forest variable importance plots based on the contributions of individual viruses in distinguishing cases from control mussels. Importance is based on the mean decrease in the Gini coefficient associated with each virus. Plot on the left is for the *G. angulata* dataset, while the plot on the right is for the *M. falcata* dataset. For both species, only a few viruses were substantially more important than others, while the majority of viruses had relatively equal contributions to the model.

**Figure 5 viruses-15-01719-f005:**
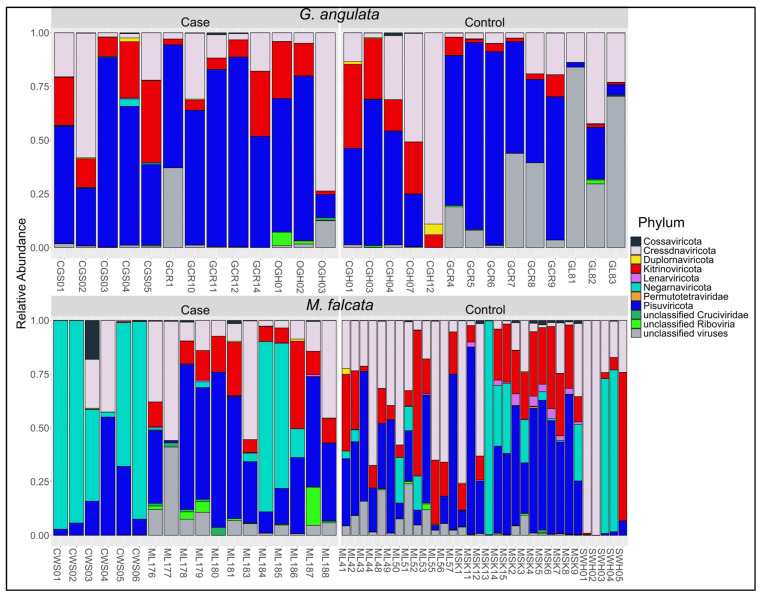
Relative viral read abundance plots at the Phylum level for *G. angulata* and *M. falcata* separated by cases and controls. Each label on the X-axis corresponds to the individual sample IDs in Appendix A.

**Figure 6 viruses-15-01719-f006:**
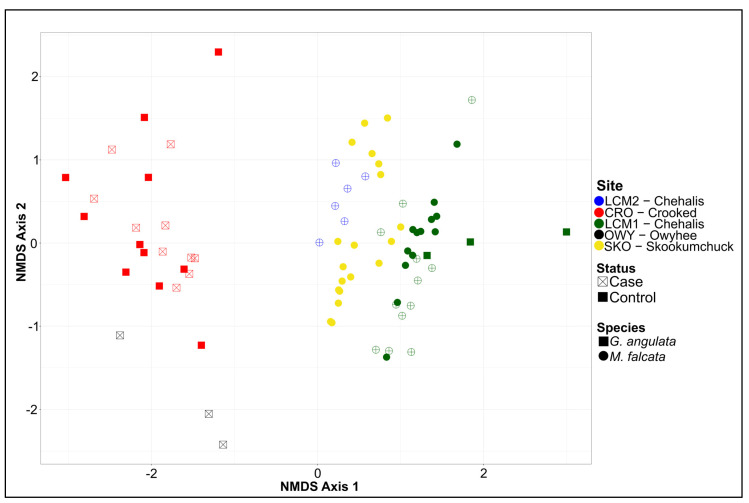
NMDS plot showing all samples used in this study. *G. angulata* are depicted as squares and *M. falcata* are depicted as circles. Control samples are shown in full color, while cases are shown as hollow shapes with crosses. Sampling sites are depicted by color, as indicated in the legend.

**Figure 7 viruses-15-01719-f007:**
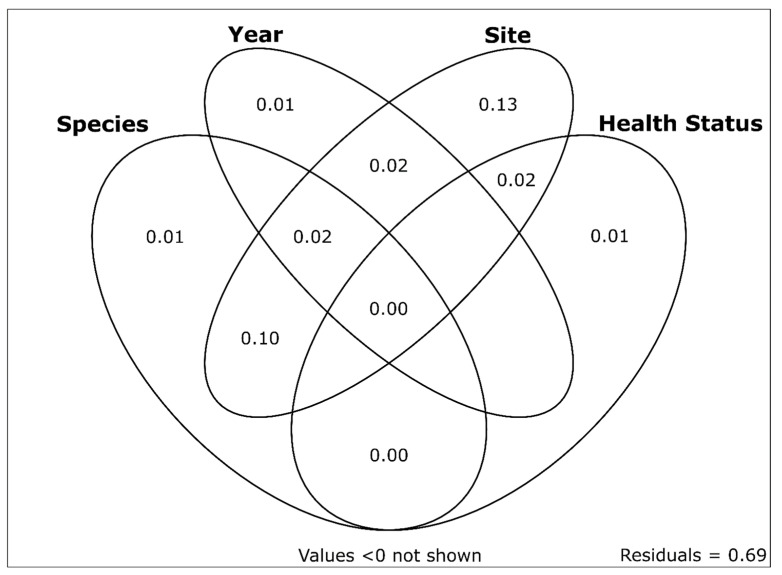
Variation partitioning plot showing the proportions of unique and shared variation in western US freshwater mussel viromes explained by sample site, sampling date, species, and health status. Numbers represent adjusted *R*^2^ values. Sample site was the only variable with a substantial proportion of unique variation explained (13.1%). Sample site and mussel species explained the only other substantial fraction of shared variation (10.2%).

**Table 1 viruses-15-01719-t001:** Sample sizes, collection dates, and locations for all study sites surveyed in this study.

River	Site Code	Site Type	Date	Species	Samples	
					Case	Control
Owyhee River	OWY	Undefined	7/25/2018	*G. angulata*	3	0
Chehalis River	LCM2	Impact	9/26/2018	*M. falcata*	6	0
Skookumchuck River	SKO	Control	9/26/2018	*M. falcata*	N/A	5
Crooked River	CRO	Impact	10/8/2018	*G. angulata*	5	5
Skookumchuck River	SKO	Control	8/24/2020	*M. falcata*	N/A	14
Chehalis River	LCM2	Impact	8/26/2020	*G. angulata*	N/A	3
	LCM2	Impact	8/26/2020	*M. falcata*	12	13
Crooked River	CRO	Impact	9/4/2020	*G. angulata*	5	6

## Data Availability

The sequence reads generated in this study are available in the NCBI Sequence Read Archive (SRA) database under BioProject accession PRJNA986535. Viral sequences described in this study have been deposited in GenBank under accession numbers OR270176-OR270732. Details for all individual mussel samples, viruses described in this study, virus richness, viral read depths, and statistical tests are provided in Appendix A.

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
