# Peer review of "Viruses of Freshwater Mussels during Mass Mortality Events in Oregon and Washington, USA"

_viruses, 2023, doi:10.3390/v15081719_

Round 1

Reviewer 1 Report

In this study, the authors analyzed viromes from 77 healthy and diseased mussels from 5 sites throughout several major Northwestern US watersheds in 2018, 2020. The majority of viral sequences detected were characterized as (+)ssRNA-containing. The authors found that mean virus richness varied between populations but was low; no significant differences were found. The authors did not find any virus whose levels were significantly higher in G. angulata case compared to controls in both years. Likewise, in the data for M. falcata, no virus was significantly elevated in diseased compared to controls in all years and at all sites. Only 1/3 of the differences in mussel viromes could be explained by species, sampling site, sampling date, and health status. Overall the work is done to a high quality using statistical analysis, the methods are unquestionable, so I think the work is very valuable and important.

Line23 – …2018-2020… Probably need a comma as there was no sampling in 2019.

Line100 – here and hereafter, the text is at page width.

Line103 – dear authors, I don't see this site on the map. It's probably a "CRO".

Line103 – you don't provide a reference, has that fact been documented?

Line108 – the abbreviation has already been deciphered earlier in the text.

Line128 - the abbreviation has already been deciphered earlier in the text.

Line184 – perhaps we should write it as 10,000 × g and similarly line186.

Line187 – Did you treat the samples with DNAase?

Line 197 -  ≥Q30 … Missing space. Q30

Line199 - … <50 nt… Missing space. < 50 nt

Line200 – So you had a control with water to remove lab contamination and it went through all stages of extraction?

Line202 – Here the program version seems to be "v", above the text for CLC is written as "version".

Line203 – Could you clarify please. What is the explanation for clustering at > 97%?

Line203-210 It's a little unclear for me. Did you first pre-identify representative sequences using blastn and blastx and then those that had E-values < 10-20 and were not assigned to eukaryotes, bacteria and bacteriophages were taken for analysis in Diamond?

Line210 I want to clarify. Did you use the input data for Diamond contigs and not ORF?

Line206 – What is the explanation for choosing this particular value (E-values < 10-20)?

Line210 – What Diamond parameters did you use?

Line225 – I think you should decipher RNA-dependent RNA polymerase.

Line230 – Please specify the program you used to mapping the reads.

Line531 – Please provide the authors' initials. J. Raghwani, S. Campbell etc.

Line575 – In Figure S1 the names of taxa are very small and unreadable, please enlarge.

1.       Could you please tell me if you have tried to use the database IMG/VR or RVMT https://riboviria.org/about/  (doi:10.1016/j.cell.2022.08.023)?

2.      Indeed, as the authors write, it would be interesting to monitor mussel viromes over a longer period of time during the year.

3.      Did you document physic-chemical parameters at the sampling sites?

4.      I did not see any description of what is above the sampling points. Perhaps factories, sewage treatment plants. For example, such a case was described in Adriaenssens et al. https://doi.org/10.1016/j.watres.2021.117568

5.      Of course, it would be good to analyze prokaryotic and eukaryotic sequences, besides viral sequences, in order to exclude the mass mortality of mussels from biotic factors. But this will probably be included in future studies.

Author Response

Reviewer 1

In this study, the authors analyzed viromes from 77 healthy and diseased mussels from 5 sites throughout several major Northwestern US watersheds in 2018, 2020. The majority of viral sequences detected were characterized as (+)ssRNA-containing. The authors found that mean virus richness varied between populations but was low; no significant differences were found. The authors did not find any virus whose levels were significantly higher in G. angulata case compared to controls in both years. Likewise, in the data for M. falcata, no virus was significantly elevated in diseased compared to controls in all years and at all sites. Only 1/3 of the differences in mussel viromes could be explained by species, sampling site, sampling date, and health status. Overall the work is done to a high quality using statistical analysis, the methods are unquestionable, so I think the work is very valuable and important.

  1. Line23 – …2018-2020… Probably need a comma as there was no sampling in 2019.

We have changed the text on line 23 to “2018 and 2020” to make this clearer.

  1. Line100 – here and hereafter, the text is at page width.

Thank you for pointing this out. We have corrected the formatting of the sections beginning at line 100 match the formatting of the previous sections with “Justified; body text” in the alignment settings.

  1. Line103 – dear authors, I don't see this site on the map. It's probably a "CRO".

Thank you for noting this – we have corrected the references to the “SRO” sample site by replacing them with “CRO” throughout the text to match the supplemental tables and map labeling.

  1. Line103 – you don't provide a reference, has that fact been documented?

We have updated line 105 to add a reference to the Xerces Society petition to the US Fish and Wildlife Service to list the Western ridged mussel as Endangered, which includes documentation regarding the described MMEs.

  1. Line108 – the abbreviation has already been deciphered earlier in the text.

Thank you, as noted above in the comment/response to point 3, the “SRO” label has been changed to CRO. The CRO references throughout this section are no longer abbreviations; now they are references to the specific sampling sites and are repeated to provide readers with clarity about the sampling design.

  1. Line128 - the abbreviation has already been deciphered earlier in the text.

Thank you for noting this, we have removed this redundancy.

  1. Line184 – perhaps we should write it as 10,000 × g and similarly line186.

Thank you for the suggestion – we have added spaces as requested to both instances.

  1. Line187 – Did you treat the samples with DNAase?

Yes, we treated samples with nucleases as described in the provided referenced publication for the full methods. 

  1. Line 197 -  … ≥Q30 … Missing space. ≥ Q30

Thank you for catching this. We have added the space as suggested above in line 199.

  1. Line199 - … <50 nt… Missing space. < 50 nt

Thank you for catching this. We have added the space as suggested above in line 200.

  1. Line200 – So you had a control with water to remove lab contamination and it went through all stages of extraction?

We appreciate the reviewer clarifying this point. To address this point (similar to that of Reviewer 2’s point 12), we have added additional text at lines 201-203, to clarify that we did not remove laboratory contaminants based on individual water control samples. Rather, we used a read mapping approach remove contamination based on known contaminants in public databases as well as our own in-house database of known laboratory contaminants, as determined from blank samples run regularly.

  1. Line202 – Here the program version seems to be "v", above the text for CLC is written as "version".

We thank the reviewer for noting this discrepancy. We have changed the text at line 200 to “v23.0.2” to remain consistent with our version references

  1. Line203 – Could you clarify please. What is the explanation for clustering at > 97%?

We appreciate the reviewer bringing attention to this point of clarification. We used this threshold to be consistent with our methods in previous publications, where we chose 97% as an optimum setting for clustering identical and nearly-identical sequences without over-clustering biologically meaningful strains/variants together. We now include a referenced statement referencing this previous use of the threshold at line 207.

  1. Line203-210 It's a little unclear for me. Did you first pre-identify representative sequences using blastn and blastx and then those that had E-values < 10-20and were not assigned to eukaryotes, bacteria and bacteriophages were taken for analysis in Diamond?

We thank the reviewer for raising this concern. We clustered identical and nearly-identical sequences into representative contigs, then used blastn and blastx to query those contigs in search of viruses. We then annotated the sequences identified as viral and used them for the Diamond analysis. To clarify this point, we have changed the phrasing in line 205 to make it clear that we clustered assembled contigs with CD-HIT-EST prior to BLAST analysis.

  1. Line210 I want to clarify. Did you use the input data for Diamond contigs and not ORF?

The reviewer is indeed correct. We used the contigs identified as viruses from the BLAST analysis as input into Diamond BlastX analysis, rather than BlastP of translated ORFs. We have clarified this point on line 215.

  1. Line206 – What is the explanation for choosing this particular value (E-values < 10-20)?

This is an excellent question. We selected this E-value threshold to maximize analytic specificity.   In other words, this threshold would identify true viruses and minimize false positive viral hits. In our previous work, we have encountered this problem (particularly for parvoviruses and CRESS viruses). We have added a statement to this effect on line 210.

  1. Line210 – What Diamond parameters did you use?

We used the default parameters for Diamond BlastX searches of the nr database. We have clarified this point by adding additional text at line 216.

  1. Line225 – I think you should decipher RNA-dependent RNA polymerase.

We agree with the reviewer, and have made this correction at line 230-231.

  1. Line230 – Please specify the program you used to mapping the reads.

We appreciate the reviewer catching this omission. To clarify this point, we have added text at lines 236-237 indicating that we used the “map reads to reference” function in CLC Genomics Workbench.

  1. Line531 – Please provide the authors' initials. J. Raghwani, S. Campbell etc.

Thanks for the suggestion. This appears not to be in keeping with the journal’s formatting requirements. We will therefore leave this comment to the discretion of the editor.

  1. Line575 – In Figure S1 the names of taxa are very small and unreadable, please enlarge.

We agree, sorry about that! The MEGAN outputs did not allow for control of the exported font size. We have now revised the figure to create new labels with an appropriate font size.

  1. Could you please tell me if you have tried to use the database IMG/VR or RVMT https://riboviria.org/about/  (doi:10.1016/j.cell.2022.08.023)?

We appreciate the reviewer’s suggestion. We have tried these and the many other published databases during the optimization phase of our methods development.  In our experience, and in our previous publications, the methods outlined herein outperform these databases (fewer false negative identification). We will re-explore these and other databases in the future, however.

  1. Indeed, as the authors write, it would be interesting to monitor mussel viromes over a longer period of time during the year.

We are grateful to the reviewer for their support of this point. Indeed, we are currently conducting parallel studies in other systems to begin assessing virome variation over time in freshwater mussels.

  1. Did you document physic-chemical parameters at the sampling sites?

The reviewer has raised an important point. We were not able to collect site-specific data on physicochemical parameters during this study. Due to funding constraints and COVID restrictions during some of the field work, we were not able to collect such detailed data, though we agree that it is a critical aspect for future studies. Fortunately, our collaborators are collecting just such data now for ongoing projects.

  1. I did not see any description of what is above the sampling points. Perhaps factories, sewage treatment plants. For example, such a case was described in Adriaenssens et al. https://doi.org/10.1016/j.watres.2021.117568

This is a great point. It is likely that stressors from the surrounding landscape are playing a role in these events and warrant further investigation. We have therefore added a description of a just-published study documenting exactly these types of effects to the Discussion (lines 555-557).

  1. Of course, it would be good to analyze prokaryotic and eukaryotic sequences, besides viral sequences, in order to exclude the mass mortality of mussels from biotic factors. But this will probably be included in future studies.

The reviewer is indeed correct. We are working on these analyses now and hope to publish them within the next year.

Reviewer 2 Report

The manuscript covers the actual problem of the mass extinction of the mussels Margaritifera falcata and Gonidea angulata in the rivers of the Western USA. The authors performed a metagenomic analysis of mollusk samples to elucidate the possible involvement of viruses in their mortality. After reviewing the study, some questions and comments arise that will help the authors improve the text of the manuscript and the overall impression/perception of their results:

Have you carried out histological studies of the tissues of the affected individuals?

L 16: “due to widespread” – How does widespread contribute to extinction? Rewrite this phrase.

L 20: “multi-year” – Remove this from the text or replace it with “two years”, here and below.

L 23: "in 2018-2020" – Replace with "in 2018 and 2020" here and below.

L 24-26: “We also characterized the distribution and diversity of a previously identified Novirhabdovirus related to pathogenic finfish viruses” – It is not clear from this sentence how this virus is relevant to this study.

L 58-60: “Recent work has found associations between freshwater mussel MMEs and viruses [10,11] and bacteria [12–14]” – Here it is appropriate to clarify where these studies were carried out.

L 73-75: Specify which "infectious diseases" and add a reference.

In sections 2.1.1. and 2.1.2. you have described preliminary observations that are not related to sampling for the present study. A false understanding and confusion is created in the dates and sites of observations and sampling. Merge sections 2.1.1. and 2.2.1 as well as 2.1.2. and 2.2.2 and reduce redundant monitoring information at non-sampling locations (including at L 116-121). Separate information about preliminary observations and sampling more clearly.

L 101-105: Add a reference if possible.

L 103: "SRO" hereinafter means "CRO" (as noted in the figure)?

Figure 1: Complete the figure, in addition to the sampling sites, with information about the species, the number of samples and, if possible, the date of sampling and the number of "case" and "control" individuals for greater clarity. Or create a separate table in the main text.

L 199-200: "We then used an in-house database to filter reads to mask low-complexity regions and remove eukaryotic reads and laboratory contaminants." – Write in more detail what you did.

L 203-205: Specify what database was used for BLASTn and BLASTx searches?

L 203-215: As follows from the description, the detection of viral genomes was based on similarity to known genomes by BLAST analysis. I strongly recommend using other programs (VirSorter 2 or similar) to uncover a wider range of virus sequences, including those that have no analogues in databases.

L 283: Explain here why you chose this particular virus for a more detailed analysis and why the L gene.

L 302-303: Add a reference.

L 302-303: "Another 16 viruses met our thresholds for using previously-established virus names" – Rewrite for clarity.

L 319: "were unclassified" – Do you mean at the family level? Specify.

L 381: “Samples OGH01 and OGH02…” – Specify what these samples are? It is better that it is clear from the text without accessing to the Supplementary Material, especially since the reference to the Supplementary Table is not provided here.

L 454: "which are largely confounded given the sampling design" – What do you mean? Where is NMDS axis 2 in the Figure 6?

Figure 5: Add explanations to the X-axis symbols (including a link to Supplementary Tables that list these symbols).

Move Supplementary Figure 2 to the main text.

L 93, 511: "CHMFV-1" – Why do you use an additional designation for this virus?

Create a table with the designations of the viruses in this and the previous study, this will greatly facilitate the perception of the results and eliminate possible confusion (for example, in Lines 303-306 it says "viruses we identified in previous studies of eastern freshwater mussel mortality events", but the viruses with the designations from this study, "crogonang" and "chemarfal" are given).

The results of the analysis of chemarfal virus 1 are poorly reflected, and limited to three lines of text (L 344-346). Add illustrative material and discussion of these results.

Author Response

Reviewer 2

The manuscript covers the actual problem of the mass extinction of the mussels Margaritifera falcata and Gonidea angulata in the rivers of the Western USA. The authors performed a metagenomic analysis of mollusk samples to elucidate the possible involvement of viruses in their mortality. After reviewing the study, some questions and comments arise that will help the authors improve the text of the manuscript and the overall impression/perception of their results:

  1. Have you carried out histological studies of the tissues of the affected individuals?

This is an important question raised by the reviewer. Because the samples collected in 2020 were non-lethal hemolymph samples, we were unable to conduct histopathology. The 2018 samples were preserved in a manner that would allow for histopathology, but have not yet been processed. We have added a statement about the importance of this and other sources of data to the Discussion (lines 561-563)

  1. L 16: “due to widespread” – How does widespread contribute to extinction? Rewrite this phrase.

We thank the reviewer for raising this point of clarification. We have removed the term “widespread” from line 16 to avoid any ambiguity.

  1. L 20: “multi-year” – Remove this from the text or replace it with “two years”, here and below.

This is a good clarification, thank you. We have changed this term to “two-year” at Lines 19 and 88 to make this clarification.

  1. L 23: "in 2018-2020" – Replace with "in 2018 and 2020" here and below.

We have changed line 23 to read “in 2018 and 2020” to make this clearer, as recommended here and by Reviewer 1 in point 1.

  1. L 24-26: “We also characterized the distribution and diversity of a previously identified Novirhabdovirusrelated to pathogenic finfish viruses” – It is not clear from this sentence how this virus is relevant to this study.

We appreciate the reviewer raising this point. While the abstract does not provide the full context for this virus, we do so in the introduction, along with the reference to the paper characterizing the virus itself. We felt it was important to mention the virus in the abstract, but also to focus more on covering all aspects of the analysis. We have modified the text at line 25 to indicate that this is a previously identified mussel Gammarhabdovirus.

  1. L 58-60: “Recent work has found associations between freshwater mussel MMEs and viruses [10,11] and bacteria [12–14]” – Here it is appropriate to clarify where these studies were carried out.

We appreciate the feedback from the reviewer to help improve this point. We have added text at lines 59-60 to show that these were from freshwater mussel populations of the Eastern United States.

  1. L 73-75: Specify which "infectious diseases" and add a reference.

The reviewer raises a good point. We have added clarification and additional references at lines 75-76 indicating that we are referring to the studies referenced in the previous paragraph of viruses and bacteria associated with mussel MMEs in Eastern US populations.

  1. In sections 2.1.1. and 2.1.2. you have described preliminary observations that are not related to sampling for the present study. A false understanding and confusion is created in the dates and sites of observations and sampling. Merge sections 2.1.1. and 2.2.1 as well as 2.1.2. and 2.2.2 and reduce redundant monitoring information at non-sampling locations (including at L 116-121). Separate information about preliminary observations and sampling more clearly.

We thank the reviewer for their careful consideration and thoughtful feedback on these sections. We have made changes as described above and below to clarify details of dates and sites. To help clarify the study design even further, we have added Table 1 to the main text.

  1. L 101-105: Add a reference if possible.

We have updated line 105 to add a reference to the Xerces Society petition to the US Fish and Wildlife Service to list the Western ridged mussel as Endangered, which includes documentation regarding the described MMEs. This comment was also noted by Reviewer 1 in point 4.

  1. L 103: "SRO" hereinafter means "CRO" (as noted in the figure)?

Yes, thank you for catching this mismatch. We have corrected the “SRO” references to “CRO” to match the labeling in the supplemental tables and map labels in Figure 1.

  1. Figure 1: Complete the figure, in addition to the sampling sites, with information about the species, the number of samples and, if possible, the date of sampling and the number of "case" and "control" individuals for greater clarity. Or create a separate table in the main text.

We appreciate this suggestion to improve the clarity of the sampling scheme. To address this point, we have created a new Table 1 at line 171 per the reviewer’s recommendation.

  1. L 199-200: "We then used an in-house database to filter reads to mask low-complexity regions and remove eukaryotic reads and laboratory contaminants." – Write in more detail what you did.

We have added additional text to lines 236-237 to clarify that we used a read mapping approach to remove reads with significant similarity to known contaminants based on published databases (citations added in text) and our own in-house database of known laboratory contaminants from commercial kits.

  1. L 203-205: Specify what database was used for BLASTn and BLASTx searches?

We thank the reviewer for raising this point of clarification. In order to address this, we have added an explanation at line 209 indicating that we searched the nt and nr databases.

  1. L 203-215: As follows from the description, the detection of viral genomes was based on similarity to known genomes by BLAST analysis. I strongly recommend using other programs (VirSorter 2 or similar) to uncover a wider range of virus sequences, including those that have no analogues in databases.

The reviewer raises an excellent point. Because many of the viruses we detect are only distantly related to known viruses, alternative approaches could prove useful. We have explored multiple alternative virus discovery pipelines, including Cenote-Taker2 and VirSorter2, but we have found that they consistently underperform compared to the BLAST-based approaches described. Many of these programs were developed primarily for use in the analysis of phage, which are not the focus of this study. Indeed, VirSorter was designed for the detection of prophages integrated into microbial genomes (PMCID: PMC4451026) and would therefore not have been appropriate for our study.

  1. L 283: Explain here why you chose this particular virus for a more detailed analysis and why the L gene.

We appreciate the reviewer’s feedback on this point. This particular virus was recently the subject of a detailed publication, which we cite. The L gene is the standard for phylogenetic analysis of members of the family Rhabdoviridae. To clarify this point, we have added text and the proper reference to lines 289-291 in the text.

  1. L 302-303: Add a reference.

We have added references as request at line 309.

  1. L 302-303: "Another 16 viruses met our thresholds for using previously-established virus names" – Rewrite for clarity.

We are grateful for the reviewer’s suggestion for increased clarity. To address this, we have added additional text at lines 314-315 describing the specific threshold criteria described in the methods for these instances.

  1. L 319: "were unclassified" – Do you mean at the family level? Specify.

We thank the reviewer for raising this point of clarification. To address this, we have added text at line 328 indicating that we are referring to viruses that were not classified below the Order level of Picornavirales.

  1. L 381: “Samples OGH01 and OGH02…” – Specify what these samples are? It is better that it is clear from the text without accessing to the Supplementary Material, especially since the reference to the Supplementary Table is not provided here.

We appreciate the feedback on clarity regarding sample numbers. To make the nature of these samples clearer, we have added a statement as requested at lines 387-388.

  1. L 454: "which are largely confounded given the sampling design" – What do you mean? Where is NMDS axis 2 in the Figure 6?

We thank the reviewer for pointing out this ambiguity. Our intent was to convey that because most of our sites sampled one species or the other (and only one site had samples of both species), our statistics related to sample species and site cannot be effectively disambiguated. To clarify this point, we have added additional text at lines 462-463 to clarify our meaning.

And, great catch, thank you! We have updated Figure 6 to correct the typo with NMDS Axis 2. The plot was correct as shown, but the label was incorrect.

  1. Figure 5: Add explanations to the X-axis symbols (including a link to Supplementary Tables that list these symbols).

We have added text to the figure 5 legend clarifying that the x-axis labels correspond to the sample IDs in Supplementary Table S1.

  1. Move Supplementary Figure 2 to the main text.

We have moved Supplementary Figure 2 to the main text as Figure 7.

  1. L 93, 511: "CHMFV-1" – Why do you use an additional designation for this virus?

We used the shorthand designation for this virus as it was established in the previous publication describing the virus, while none of the viruses in this manuscript have been formally classified or assigned shorthand designations. We explain this rationale at line 94.

  1. Create a table with the designations of the viruses in this and the previous study, this will greatly facilitate the perception of the results and eliminate possible confusion (for example, in Lines 303-306 it says "viruses we identified in previous studies of eastern freshwater mussel mortality events", but the viruses with the designations from this study, "crogonang" and "chemarfal" are given).

We appreciate the reviewer for raising this point of clarification. We agree that the nomenclature can be confusing. Fortunately, a paper was just published by member of the International Committee on the Taxonomy of Viruses outlining rules for public database submission of uncultivated virus genome sequences for taxonomic classification (https://www.ncbi.nlm.nih.gov/pubmed/37430074). We have followed these guidelines and now explain that we did so and reference this paper (line228). We have also modified lines 314-315 to clarify that our threshold for using previously-established names refers to complete RdRP/Rep sequences.

  1. The results of the analysis of chemarfal virus 1 are poorly reflected, and limited to three lines of text (L 344-346). Add illustrative material and discussion of these results.

We appreciate the reviewer’s feedback on this point. Because the sequences were nearly identical, we omitted phylogenetic trees in favor of simply stating the sequence identity percentages to brevity. To ensure that these data are placed in context, we have included a paragraph interpreting the results and implications of our findings (lines 525-538).

Reviewer 3 Report

The paper describes a search for a potentially unknown virus that may be killing freshwater mussels.  Although a pathogen was not identified that caused these mortalities and morbidities, the paper describes identification of numerous unknown virus. In its own right, it will make an important contribution to the literature. The paper is well written. I recommend its publication as is

Author Response

Reviewer 3

The paper describes a search for a potentially unknown virus that may be killing freshwater mussels.  Although a pathogen was not identified that caused these mortalities and morbidities, the paper describes identification of numerous unknown virus. In its own right, it will make an important contribution to the literature. The paper is well written. I recommend its publication as is

We thank the reviewer for their kind feedback and consideration of our manuscript!

Round 2

Reviewer 2 Report

The authors answered the main questions and made some clarifications in accordance with my comments.